# Regional Variability of Raindrop Size Distribution from a Network of Disdrometers over Complex Terrain in Southern China

**Asi Zhang [1], Chao Chen [1,2,*] and Lin Wu [1]**

[1] Guangdong Meteorological Observatory, Guangzhou 510640, China
[2] State Key Laboratory of Severe Weather, Chinese Academy of Meteorological Sciences, Beijing 100081, China
[*] Correspondence: chenchao@gd121.cn

**Abstract:** Raindrop size distribution (DSD) over the complex terrain of Guangdong Province, southern China, was studied using six disdrometers operated by the Guangdong Meteorology Service during the period 1 March 2018 to 30 August 2022 (~5 years). To analyze the long-term DSD characteristics over complex topography in southern China, three stations on the windward side, Haifeng, Enping and Qingyuan, and three stations on the leeward side, Meixian, Luoding and Xuwen, were utilized. The median mass-weighted diameter ($D_m$) value was higher on the windward than on the leeward side, and the windward-side stations also showed greater $D_m$ variability. With regard to the median generalized intercept ($\log_{10} N_w$) value, the $\log_{10} N_w$ values decreased from coastal to mountainous areas. Although there were some differences in $D_m$, $\log_{10} N_w$ and liquid water content (*LWC*) frequency between the six stations, there were still some similarities, with the $D_m$, $\log_{10} N_w$ and *LWC* frequency all showing a single-peak curve. In addition, the diurnal variation of the mean $\log_{10} N_w$ had a negative relationship with $D_m$ diurnal variation although the inverse relationship was not particularly evident at the Haifeng site. The diurnal mean rainfall rate also peaked in the afternoon and exceeded the maximum at night which indicated that strong land heating in the daytime significantly influenced the local DSD variation. What is more, the number concentration of drops, $N(D)$, showed an exponential shape which decreased monotonically for all rainfall rate types at the six observation sites, and an increase in diameter caused by increases in the rainfall rate was also noticeable. As the rainfall rate increased, the $N(D)$ for sites on the windward side (i.e., Haifeng, Enping and Qingyuan) were higher than for the sites on the leeward side (i.e., Meixian, Luoding and Xuwen), and the difference between them also became distinct. The abovementioned DSD characteristic differences also showed appreciable variability in convective precipitation between stations on the leeward side (i.e., Meixian, Luoding and Xuwen) and those on the windward side (Haifeng and Enping, but not Qingyuan). This study enhances the precision of numerical weather forecast models in predicting precipitation and verifies the accuracy of measuring precipitation through remote sensing instruments, including weather radars located on the ground.

**Keywords:** regional variability; raindrop size distribution; complex terrain

## 1. Introduction

Cloud and precipitation microphysical processes are core components of atmospheric water cycles [1,2] and key factors related to the heat, water vapor and momentum balance of the Earth's atmosphere [3,4]. They affect not only local and short-term weather processes, but also atmospheric circulation and global climate change [5]. The condensation, evaporation and sublimation of cloud and precipitation particles govern the evolution of raindrop size distribution (DSD) which is the fundamental characterization of rain microphysics.

Knowledge of DSD and its temporal and spatial variability is critical to understanding the characteristics of precipitation system microphysics over complex terrain, developing

better quantitative precipitation estimation (QPE) algorithms for rainfall and improving parameterization in numerical weather prediction models [6,7], because DSD variations affect the mass flux or rainfall rate around the surface [8]. Furthermore, numerical weather prediction models can in turn improve simulations of precipitation through better characterization of microphysical development using model parameterization schemes. Numerous studies have been carried out concerning DSD variations associated with different seasons [9,10], weather systems (e.g., frontal precipitation [11,12], mesoscale convective systems [13–15], tropical cyclones [16,17]), geographical locations [18,19] and types of precipitation [20,21].

Because DSD varies by location and climate regime, it is important to investigate the DSD in a specific area to improve understanding of the local evolution and variability of DSD. What is more, the land topography and ocean can also affect the local DSD which causes the DSD near the land–sea boundary to have more variability compared to the inland and oceanic regions separately [22]. The intricate topography in southern China can generate intense rain because of the interplay of the monsoon flow and local circulations caused by the terrain. The locations of heavy rainfall may differ occasionally when a comparable southwesterly or nearly stationary subtropical front is present. Heavy rainfall can transpire over mountainous areas, slopes, and coastal regions. Studies show that the mean rainfall intensity differs on the leeward side and windward side [23,24]. In addition, winds originating from the oceanic area carry extra moisture, resulting in increased specific humidity over southern China and creating favorable conditions for vigorous convection [25].

Many researchers have conducted DSD measurements in China. Using nine months of DSD observations from ten disdrometers, Han et al. [26] showed that Beijing, located in north China, had lower $D_m$ and $N_w$ values compared to south China, which they attributed to the influence of terrain. Liu et al. [27] studied the temporal and spatial variabilities utilizing 34 disdrometers over a two-year period in southwest China, and found significantly different DSD characteristics and parameter relationships between southwest China and other regions such as north, eastern and southern China. Based on use of a disdrometer over four months, Wu et al. [28] observed that the number concentration for larger raindrops in southern China exceeded that of the Tibetan Plateau. The lower convective frequency in the Tibetan Plateau can be attributed to the lower humidity and higher altitude, as well as to the differences in geographic location and climate of these two locations. In addition to studies using DSD observation data from different sites in China, Zeng et al. [29] examined the DSD of monsoon seasons over the South China Sea (SCS) using disdrometer data collected during marine surveys from 2012 to 2016 and found significant variations in raindrop concentration between pre-monsoon, monsoon and post-monsoon periods over the SCS.

It can be seen that existing DSD investigations have been based primarily on observation measurements of a short duration at relatively uniform terrain. In this study, the observed DSD data, extending over several years and covering several locations, such as inland and coastal areas on the windward and leeward sides, is still fairly limited. Although the characteristics of the microphysical processes at different locations are difficult to represent because of their large temporal and spatial variability, nearly five years' worth of disdrometer observations can help to improve our understanding of the properties of cloud droplets and precipitation types such as convective and stratiform rain regimes. This paper utilizes approximately five years of disdrometer data gathered between 1 March 2018 and 30 August 2022 in order to characterize DSD variabilities in complex terrain, and is organized as follows: details of the study area, datasets and methodology are presented in Section 2; the characteristics of DSD at the six different sites are reported in Section 3; and a summary and conclusions are provided in Section 4.

## 2. Study Area, Data and Methodology

### 2.1. Study Area

Guangdong Province is located in southern China and is influenced by the East Asian summer monsoon. The terrain is complex, with the meandering Nanling mountains in the north and the vast South China Sea to the south. The terrain in the province is generally high in the north and low in the south (see Figure 1) which provides an ideal test bed for understanding how DSD characteristics are influenced by complex topography. Affected by land–sea contrast [30], low-level jets [31], orographic lifting [32] and monsoon flow [33], the variability of precipitation in this region is very complex and triggers various cloud microphysical processes such as collision–coalescence, aggregation, condensation and so on. In addition, Guangdong is one of the provinces of China with maximum region-averaged rainfall accumulation because of the more frequent occurrence of precipitation systems [34]. However, the temporal and spatial distribution of precipitation is uneven. Accordingly, located on the windward slope of Guangdong Province, Haifeng, Enping and Qingyuan experience 1600~2200 mm of rainfall during the monsoon season whereas, located on the leeward side, Meixian Luoding and Xuwen experience only 1000~1500 mm.

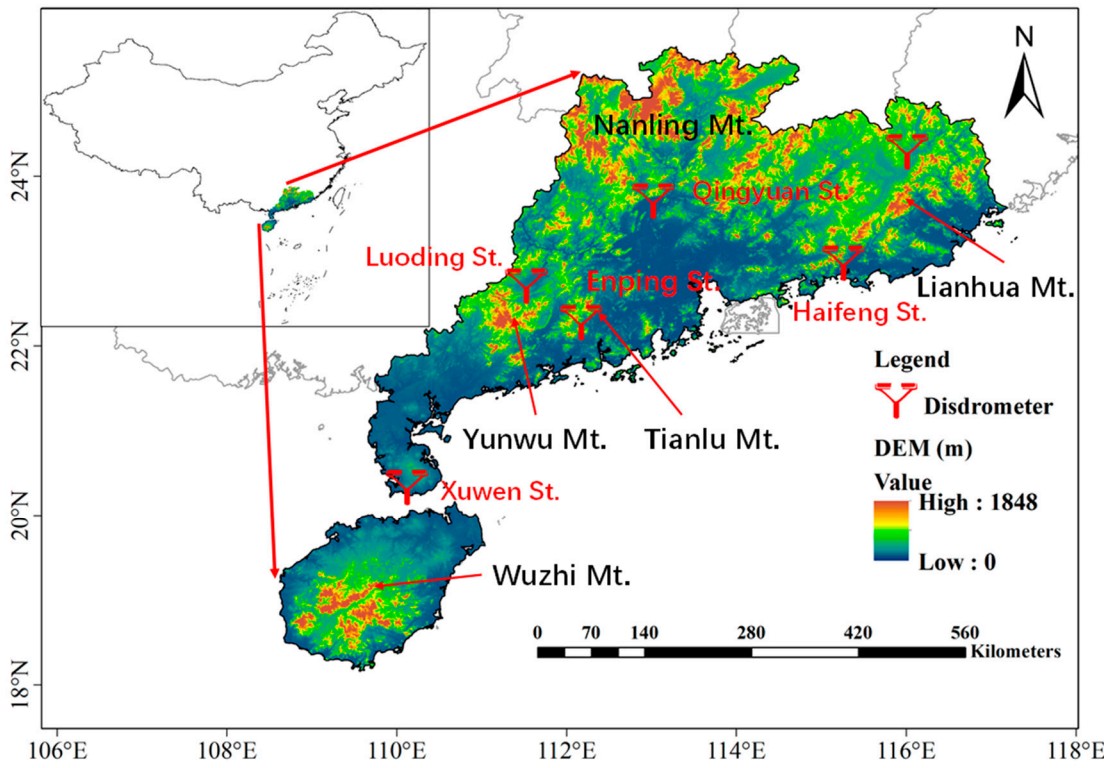

**Figure 1.** Guangdong province and its digital elevation model (DEM). The red symbols are the disdrometer locations used in this study. The locations of Nanling, Lianhua, Tianlu, Yunwu and Wuzhi Mountains are also displayed.

### 2.2. Data and Methodology

The datasets used in this study are collected from part of a network of HY-P1000 disdrometers (hereinafter collectively referred to as the disdrometers) distributed in Guangdong Province and monitored by the Guangdong Meteorological Service. All disdrometers in the province are of the same type: the HY-P1000 disdrometer, manufactured by Huayun Technology Development Corporation, China. This network of disdrometers was initiated in 2018 and gradually the network grew to include up to 86 disdrometers distributed across Guangdong Province. In this study, six stations were selected, namely the Meixian, Haifeng, Luoding, Enping, Xuwen and Qingyuan sites, which are at the foot of the north side of the Lianhua Mountain, the south side at the Lianhua Mountain, the north side of the

Yunwu Mountain, the south side of Tianlu Mountain, the north side of Wuzhi Mountain and the south side of the Nanling Mountains, respectively. Their characteristics are listed in Table 1. Haifeng, Enping and Qingyuan are the heavy rainfall areas of Guangdong whereas Meixian, Luoding and Xuwen are among the areas with the least rainfall. The highest disdrometer used in this study is located at Meixian, at 89.3 m, and the lowest is at just 5.7 m, located at the Haifeng site.

Disdrometers [28,35,36], which operate on a similar physical principle to OTT Parsivel disdrometers [37], can measure the velocity of precipitation ranging from 0.05 to 20.8 m s$^{-1}$ and particle size ranging from 0.062 to 24.5 mm, which can both be separated into 32 classes. The lowest two channel sizes were not used because of their low signal-to-noise ratio [38]. The sample area was about 54 cm$^2$, and the sampling interval for the particle size is finer for smaller- and medium-sized particles and broadened for larger precipitation raindrops [39]. To identify and remove suspicious instrumental errors (due to splashing, strong winds, multiple raindrops at a time, marginal raindrops, insects, spiders, etc.), a filter method was used which excluded from the raw disdrometer data rainfall rates that are too low (<0.1 mm/h) or raindrops that are too large (8 mm), to reduce the sampling error. This study only focused on rainfall samples, and the mix-phased particles are excluded from the raw datasets [14]. Meanwhile, a velocity-based filter was also used to exclude the particles that are unlikely to be raindrops [40]. One minute of DSD data with a total of at least 10 raindrops during a minimum of 10 min of consecutive rainfall is used for calculation in this study. A total of about five years (2018–2022) of disdrometer data from the Meixian, Haifeng, Luoding, Enping, Xuwen and Qingyuan stations in Guangdong Province were utilized to understand the temporally continuous, natural DSD variations. During the quality control procedure, it was found that some disdrometers had data gaps, which could have been caused by device problems or power failures. However, after the quality control, a total of 63,777, 54,296, 58,170, 78,710, 34,079 and 75,370 one-minute periods of raindrop spectra were observed using the disdrometers over Meixian, Haifeng, Luoding, Enping, Xuwen and Qingyuan, respectively, during the five-year period, 2018–2022.

Following the quality control, the concentration of raindrops with diameters in the range of a unit size interval $N(D_i)$ was calculated as follows:

$$N(D_i) = \sum_{i=1}^{32} \frac{n_{ij}}{A_i \cdot \Delta t \cdot V_j \cdot \Delta D_i}, \tag{1}$$

where $n_{ij}$ is the number of raindrops in each bin; $A_i$ is the sampling area ($54 \times 10^{-4}$ m$^2$); $\Delta t$ is the sampling time (60 s); $V_j$ is the fall speed in the $j$th size bin (m s$^{-1}$); and $\Delta D_i$ is the width of the $i$th size bin (cm).

After the calculation of $N(D_i)$, the rainfall rate ($R$) and liquid water content ($LWC$) could be calculated as follows:

$$R\left(\text{mm h}^{-1}\right) = \frac{6\pi}{10^4} \sum_{i=1}^{32} \sum_{j=1}^{32} V_j \cdot N(D_i) \cdot D_i^3 \cdot \Delta D_i, \tag{2}$$

$$LWC\left(\text{g m}^{-3}\right) = \frac{\pi \rho_w}{6000} \sum_{i=1}^{32} N(D_i) \cdot D_i^3 \cdot \Delta D_i \tag{3}$$

The $n$th-order moment of the DSD is defined as

$$M_n = \int_0^\infty N(D) \cdot D^n \cdot dD, \tag{4}$$

The mass-weighted mean diameter $D_m$ (mm) is

$$D_m = \frac{M_4}{M_3}, \tag{5}$$

The normalized intercept parameter $N_w$ (m$^{-3}$ mm$^{-1}$) is expressed as

$$N_w \left( m^{-3} mm^{-1} \right) = \frac{4^4}{\pi \rho_w} \cdot \frac{10^3 LWC}{D_m^4}, \tag{6}$$

where $\rho_w$ is the density of water and *LWC* represents the liquid water content.

**Table 1.** Location, height and number of 1 min DSD from the six disdrometer sites.

| Site | Location | Altitude (m) | 1 min DSD Spectra | Location |
|------|----------|--------------|-------------------|----------|
| Meixian | 24.3°N, 116.1°E | 89.3 | 63,777 | Leeward |
| Haifeng | 23.0°N, 115.3°E | 5.7 | 54,296 | Windward |
| Luoding | 22.7°N, 111.6°E | 57.8 | 58,170 | Leeward |
| Enping | 22.3°N, 112.2°E | 25.4 | 78,710 | Windward |
| Xuwen | 20.3°N, 110.2°E | 69.0 | 34,079 | Leeward |
| Qingyuan | 23.7°N, 113.1°E | 19.4 | 75,370 | Windward |

To evaluate the accuracy of the disdrometer observations, the hourly accumulated precipitation collected by the disdrometers was compared with data from the ground rain gauges at automatic weather stations (Figure 2). Correlation coefficient (*CC*), standard deviation (*SD*) and relative bias (*RB*) were used to evaluate the performance of the disdrometer observations and the formulas are presented in Equations (7)–(9). Although there are some differences between the two measurements, in general the disdrometers are in good agreement with the rain gauges, with a high *CC* of 0.96 and low *SD* of 1.77 mm for a total of 2328 h of data. Therefore, the disdrometer measurement can be used as a reference for 1 min rainfall and DSD calculation in this study.

$$CC = \frac{\sum_{i=1}^n (x_i - \bar{x})(y_i - \bar{y})}{\sqrt{\sum_{i=1}^n (x_i - \bar{x})^2 \times \sum_{i=1}^n (y_i - \bar{y})^2}}, \tag{7}$$

$$SD = \sqrt{\frac{1}{n-1} \sum_{i=1}^n \left( (x_i - y_i) - \overline{(x-y)} \right)^2} \tag{8}$$

$$RB = \frac{1}{n} \sum_{i=1}^n \frac{x_i - y_i}{y} \tag{9}$$

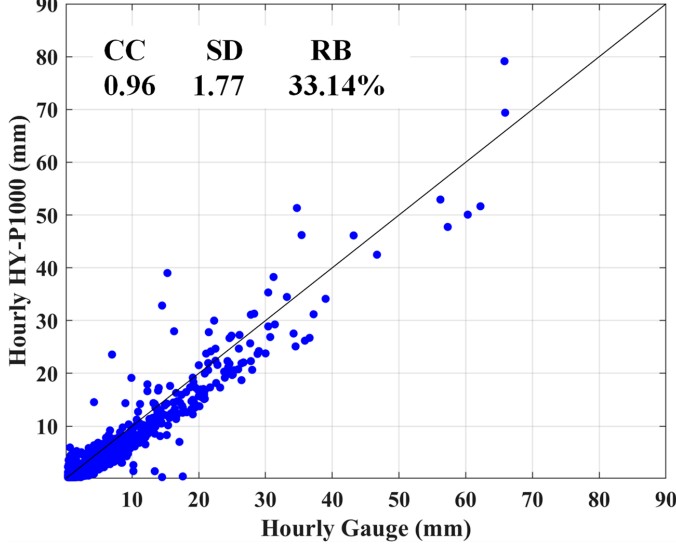

**Figure 2.** Scatter plot of hourly accumulated rainfall collected from the HY-P1000 disdrometers and rain gauges from automatic weather stations.

*2.3. Separation of Rainfall*

The one-minute DSD data from the six stations were classified into stratiform and convective precipitation by adopting the classification method proposed by Bringi et al. [41]. A brief description of the classification algorithm is as follows: for a sample of $R(t_i)$ at the instant $t_i$, if the $R$ values from $t_i - N$ to $t_i + N$ lie in the range of 0.5–5 mm h$^{-1}$ and their standard deviation is less than 1.5 mm h$^{-1}$ (N is set to be 5 samples), it is classified as stratiform precipitation; otherwise, if the values of $R$ from $t_i - N$ to $t_i + N$ are greater than 5 mm h$^{-1}$ and the standard deviation is greater than 1.5 mm h$^{-1}$, the sample is classified as convective. Table 2 lists the samples, mean $D_m$, mean $\log_{10}N_w$, mean $R$ and *LWC* of connective and stratiform following classification. To explain the DSD variations, this study classifies raindrop diameter as small (less than 1 mm), medium (between 1 and 3 mm) and large (over 3 mm), as proposed by previous studies [42,43]. As a result, the stratiform precipitation samples are much larger than the convective precipitation samples from the six sites which means that stratiform precipitation dominated over the whole year. It is obvious that the $D_m$, $\log_{10}N_w$, $R$ and *LWC* were higher for convection than for stratiform precipitation. On average, the $D_m$ of all the sites is small- to medium-sized raindrops which are smaller than 3 mm. Qingyuan, an area of heavy rainfall located far from the ocean, has the highest $D_m$ value of convection rain. The convective rainfall mean $\log_{10}N_w$ values of the six sites are in the order Luoding > Xuwen > Enping > Meixian > Haifeng > Qingyuan. In addition, Luoding has the highest $\log_{10}N_w$ which is probably related the seeder–feeder mechanism [44] that can enhance the coalescence, thereby increasing the concentration of raindrops and causing the largest $\log_{10}N_w$. On the other hand, the stratiform rainfall has a lower concentration of raindrops and *LWC*, indicating that raindrop growth is weaker in stratiform rain.

**Table 2.** Rain parameters derived from the raindrop spectra for convective (C) and stratiform (S) rain. $D_m$, $\log_{10}N_w$, $R$ and *LWC* are the mass-weighted mean diameter, generalized raindrop concentration, rainfall rate and liquid water content, respectively.

| Station | Rain Type | Sample | $D_m$ (mm) | $\log_{10}N_w$ (m$^{-3}$ mm$^{-1}$) | $R$ (mm h$^{-1}$) | $LWC$ (g m$^{-3}$) |
|---|---|---|---|---|---|---|
| Meixian | C * | 2280 | 2.18 | 3.65 | 37.43 | 1.52 |
| | S * | 9161 | 1.31 | 3.53 | 1.84 | 0.11 |
| Haifeng | C | 2245 | 2.34 | 3.61 | 44.69 | 1.79 |
| | S | 7524 | 1.32 | 3.54 | 1.86 | 0.11 |
| Luoding | C | 2351 | 2.27 | 3.74 | 52.27 | 2.11 |
| | S | 8619 | 1.37 | 3.44 | 1.87 | 0.11 |
| Enping | C | 4235 | 2.19 | 3.69 | 41.51 | 1.70 |
| | S | 10,157 | 1.45 | 3.34 | 1.94 | 0.11 |
| Xuwen | C | 1634 | 2.26 | 3.72 | 47.39 | 1.90 |
| | S | 4273 | 1.42 | 3.34 | 1.86 | 0.11 |
| Qingyuan | C | 4112 | 2.43 | 3.55 | 48.87 | 1.91 |
| | S | 10,683 | 1.43 | 3.34 | 1.60 | 0.11 |
| All | C | 2809 | 2.28 | 3.67 | 45.36 | 1.82 |
| | S | 8402 | 1.38 | 3.42 | 1.83 | 0.11 |

* C means convective precipitation, S means stratiform precipitation.

## 3. Results

*3.1. DSD Parameter Distributions*

Figure 3 shows the $D_m$ and $\log_{10}N_w$ distribution for the six sites. The $D_m$ median value distributions are all similar, at about 1.2 mm. Specifically, the $D_m$ median value is slightly lower for Meixian (1.16 mm), Luoding (1.19 mm) and Xuwen (1.24 mm) than for Haifeng (1.22 mm), Enping (1.30 mm) and Qingyuan (1.28 mm). It is evident that the median $D_m$ value is higher on the windward than on the leeward side. The $D_m$ distributions also show a relatively large variability at Haifeng, Enping and Qingyuan. The upper extremity of the $D_m$ box and whisker plot decrease, moving from Qingyuan (2.93 mm) to Haifeng (2.92 mm). The minimum value of $D_m$ is also lowest in Haifeng (0.34 mm), making the variability of

$D_m$ in this region slightly greater. For the 25th percentage value of $D_m$, Enping, Xuwen and Qingyuan show a higher value (about 0.90 mm) than Meixian, Haifeng and Luoding (about 0.82 mm). With regard to the $\log_{10}N_w$ distribution, Haifeng differs slightly from the other five sites, with the highest raindrop concentration (3.61). The $\log_{10}N_w$ distributions for Meixian, Luoding, Enping and Xuwen are almost identical, with similar median values (3.55, 3.56, 3.56 and 3.55, respectively). Qingyuan has the lowest $\log_{10}N_w$ median value of the six sites (3.48). Meixian and Xuwen have similar maximum values of 5.16 and 5.18, respectively, whereas the maximum values of the other sites (i.e., Haifeng, Luoding and Enping) are higher at 5.32, 5.32 and 5.26, respectively. Qingyuan still has the lowest maximum value of $\log_{10}N_w$ (5.07). With regard to the minimum value, Haifeng, Xuwen and Qingyuan are similar, at around 1.86. Meixian has the highest minimum value at 1.96 whereas Luoding has the lowest value at 1.80. It can be easily concluded that Meixian has the least variability of $\log_{10}N_w$ whereas Luoding has the greatest variability of $\log_{10}N_w$ even though they are both on the leeward side of mountains. The above analysis clearly shows that, moving from coastal to mountainous areas, the median $\log_{10}N_w$ values decrease. This variation of $N_w$ may be explained by distance from the ocean. The supply of water vapor near the coastal areas is much more than in the inland areas which develop condensation into droplets and small raindrops increasing the raindrop number concentration [45]. In addition, Meixian, Luoding and Xuwen, with lower annual rainfall, have a smaller $D_m$ than Haifeng, Enping and Qingyuan which are the heavy rainfall areas of Guangdong. This kind of variation can be attributed to differences in terrain [46].

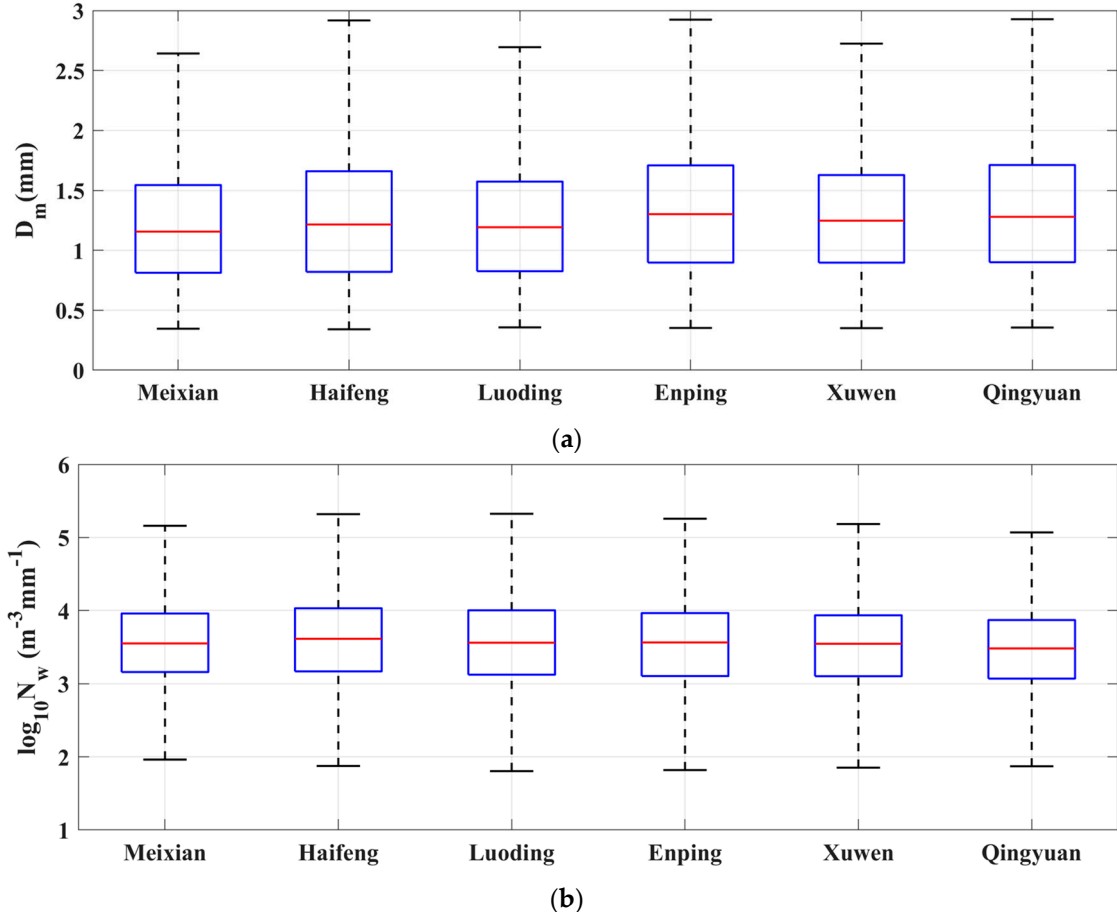

**Figure 3.** Box−and−whisker plot of (**a**) $D_m$ and (**b**) $\log_{10}N_w$ distribution over the six different sites. The box represents the data between the 25th and 75th percentiles and the whiskers show the maximum and minimum values. The horizontal red line within the box represents the median value of the distribution.

To further investigate DSD variability across the six regions, $D_m$, $\log_{10}N_w$ and *LWC* frequency distribution are shown in Figure 4. For all of the above parameters, a similar trend is evident in all six areas. From Figure 4a, it can be seen that most of the $D_m$ values at the six locations are between 0.2 mm and 2.8 mm and show a single-peak curve. Luoding, Enping and Qingyuan have a similar peak value of 1.2 mm; however, Meixian and Xuwen have a similar peak value of 1.0 mm. When the $D_m$ is less than 1.0 mm, the peak values for Meixian and Luoding are higher than those for Enping and Qingyuan which means that Meixian and Luoding have a greater number of small raindrops, indicating active drop breakup processes in those areas. When the $D_m$ is greater than 1.5 mm, the frequency at Haifeng, Enping and Qingyuan is higher than at the other locations. The $\log_{10}N_w$ frequencies for the six locations are shown in Figure 4b. When the $\log_{10}N_w$ is less than 3.7, the six locations show a similar $\log_{10}N_w$ frequency trend, with a slightly higher $\log_{10}N_w$ frequency at the Qingyuan site. When the $\log_{10}N_w$ is greater than 3.7 and less than 4.3, Xuwen has the highest $\log_{10}N_w$ frequency and Qingyuan has the lowest. The *LWC* is clearly higher at Haifeng, Enping, Xuwen and Qingyuan compared with Meixian and Luoding when the *LWC* is over 0.2 g m$^{-3}$, as shown in Figure 4c. In general, the Meixian and Luoding areas which are far from the ocean and on the leeward side of mountains have a similar *LWC* frequency trend whereas those of Haifeng, Enping, Xuwen and Qingyuan, which are close to the ocean or on the windward side of mountains, are similar to each other. Despite some minimal similarities, the DSD distribution at coastal sites and inland sites remained different, evidence of the variation cloud microphysical processes in these six regions.

The diurnal variation of the mean $D_m$ in the six locations over the approximately five-year period is shown in Figure 5a. The number variation for every hour over the Meixian, Haifeng, Luoding, Enping, Xuwen and Qingyuan areas is also displayed in Figure 6. The mean $D_m$ variation of the six locations has a diurnal cycle with a maximum in the daytime (0800–2000 LST, local standard time). Meixian and Luoding have a similar $D_m$ trend whereas the other four sites show much more variability, in particular during the daytime. The intense convective activity during the daytime, particularly in the late afternoon, can enhance the collision–coalescence process, thereby increasing $D_m$ values [43]. The coastal areas (i.e., Haifeng, Enping and Xuwen) also show a maximum value in the daytime which indicates that complex terrain also influences the microphysical processes in coastal areas.

The mean $\log_{10}N_w$ diurnal variation seems to have a negative relationship with the $D_m$ diurnal variation, although the inverse relationship is not very notable at the Haifeng site. All sites except Haifeng have a primary maximum in the morning (1000–1200 LST) and a secondary peak at night (0100–0400 LST). The $\log_{10}N_w$ has its minimum value in the late afternoon, which is probably because precipitating convective clouds are present at this time [47]. The intense convective process affects the DSD by drop-sorting and enhancing the collision coalescence process [43] which can lift the smaller drops to a higher altitude and consume the smaller drops, thus intensifying the growth of large raindrops.

It can be seen that the precipitation rates in Meixian and Luoding on leeward slopes present a single-peak structure, successively peaking in the afternoon, followed by a gradual slowing, as shown in Figure 5c. In addition to the precipitation peak in the daytime, Haifeng, Enping, Xuwen and Qingyuan, on the coast or on the windward side, also have a subpeak at night. Studies [48,49] have shown that marine precipitation peaks at night, whereas precipitation on land generally peaks in the afternoon. In this study, the three coastal sites and one windward site have a peak both during the day and at night which deviates from the typical ocean and land pattern because of the joint influence of sea and land winds over the complex coastal and inland terrain. Compared with Xuwen (10.5 mm h$^{-1}$), the peak precipitation rates at Haifeng (8.0 mm h$^{-1}$), Enping (9.0 mm h$^{-1}$) and Qingyuan (7.9 mm h$^{-1}$) at night are much weaker, which could be mainly attributed to Xuwen's unique geographical location. Xuwen is not only a coastal location but, as can be seen from the DEM map, it is also subject to the influence of the Hainan Wuzhi Mountain to the south.

In addition, it can also be observed that the daytime (0800–2000 LST) maximum of the six locations is greater than the maximum at night (2000–0800 LST). Previous studies [50,51] found that strong land heating in the daytime significantly influences the formation of deep convective rainfall from 0800 to 2000 LST, making the updrafts more intensive and producing larger raindrops. This is evident from Figure 5a,b.

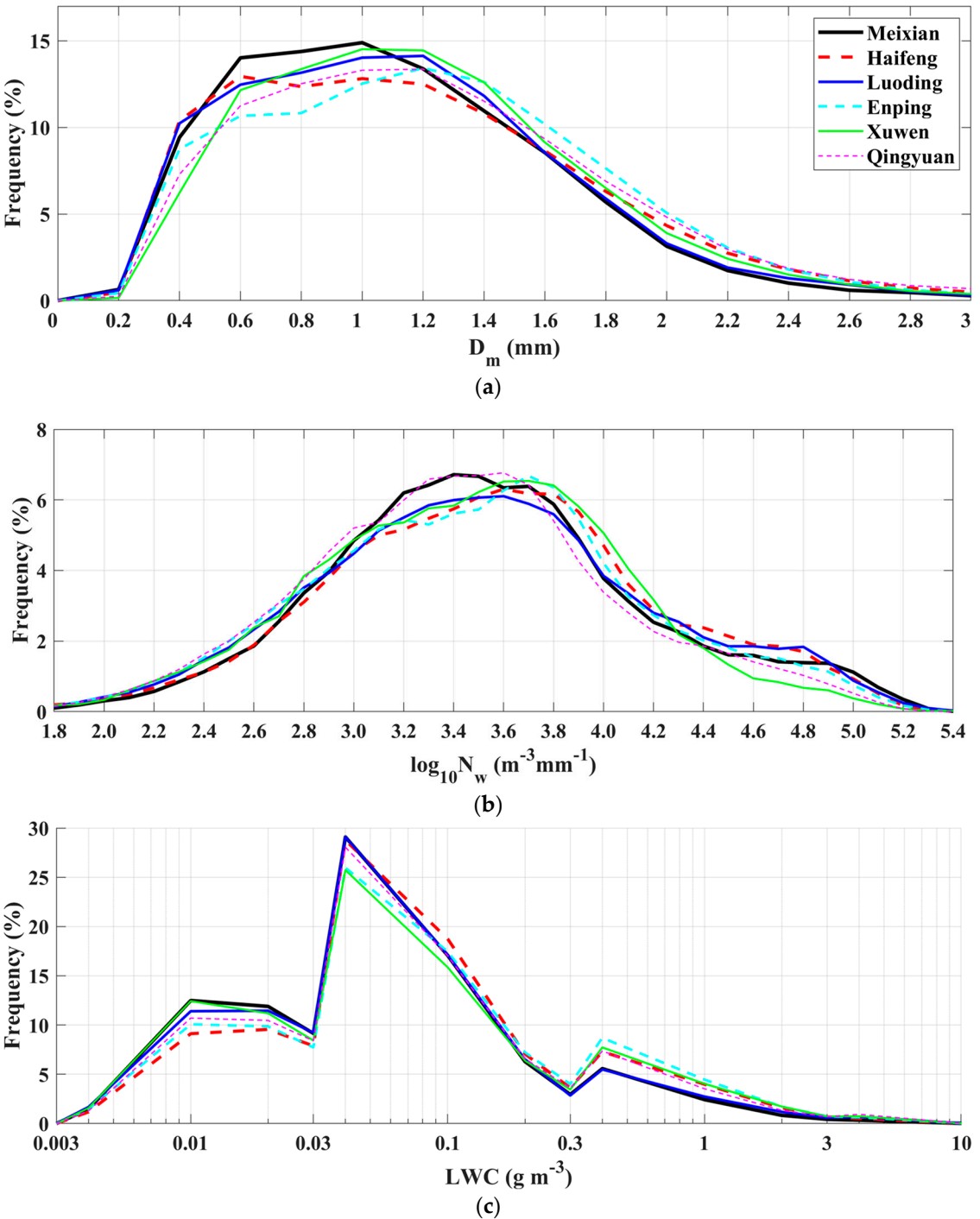

**Figure 4.** Frequency of (**a**) $D_m$, (**b**) $\log_{10}N_w$ and (**c**) *LWC* over the Meixian, Haifeng, Luoding, Enping, Xuwen and Qingyuan areas.

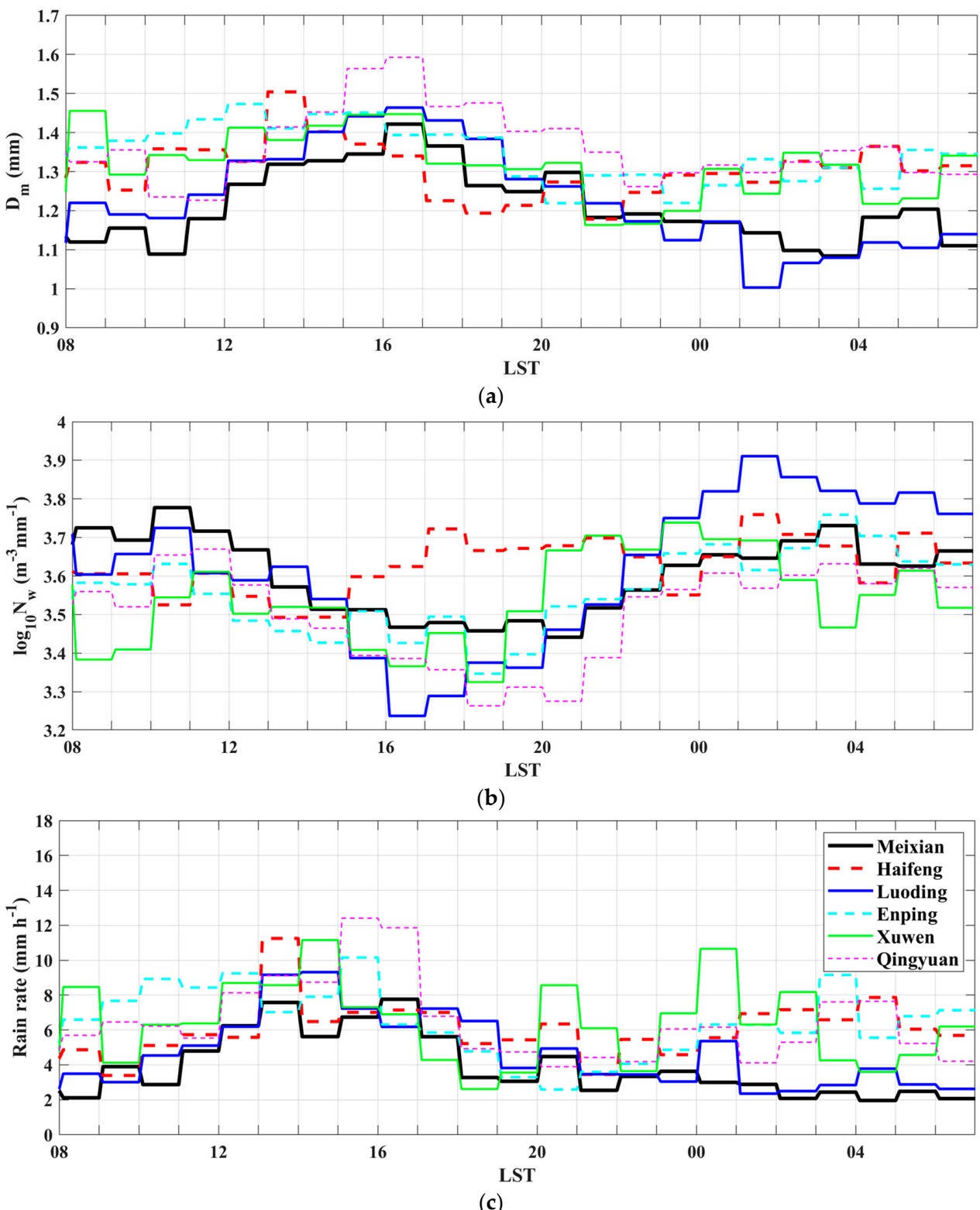

**Figure 5.** Diurnal variation of the mean (**a**) $D_m$ (**b**) $\log_{10}N_w$ and (**c**) rain rate over the Meixian, Haifeng, Luoding, Enping, Xuwen and Qingyuan areas.

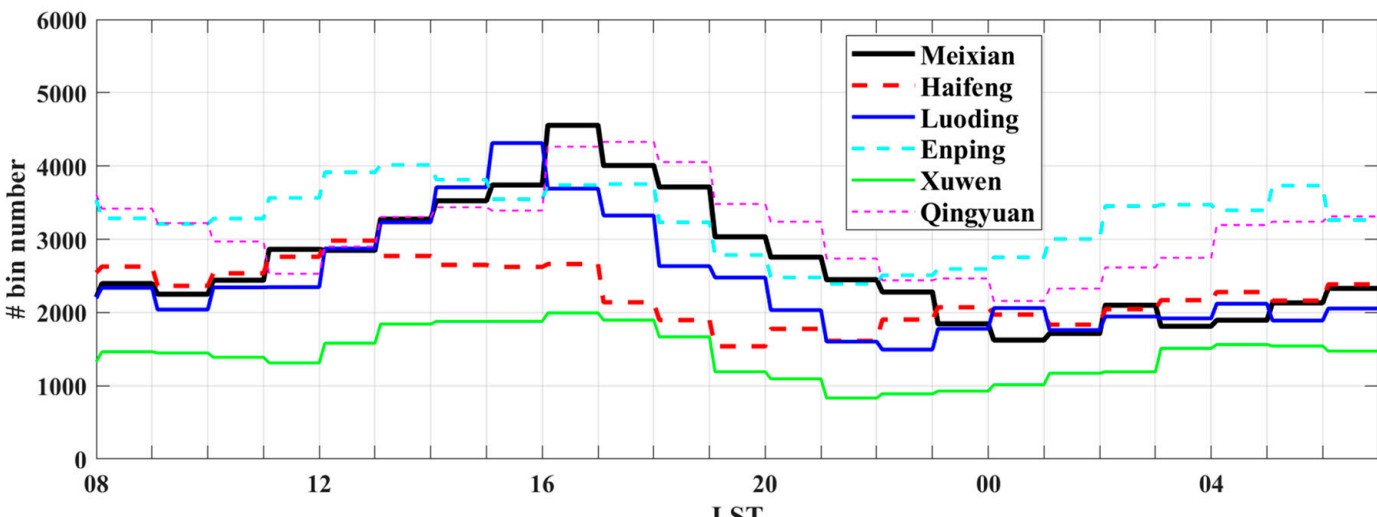

**Figure 6.** The number variation for every hour over the Meixian, Haifeng, Luoding, Enping, Xuwen and Qingyuan areas.

### 3.2. Averaged Drop Size Distributions

Figure 7 shows the average raindrop concentration and raindrop diameter *D* for Meixian, Haifeng, Luoding, Enping, Xuwen and Qingyuan from 2018 to 2022. In order to adapt the large variability of *N*(*D*), Figure 7 is plotted on a logarithmic scale. We focus on diameters below 1.0 mm to better understand the differences in concentration at small drop sizes. It was observed that the DSDs measured at one-minute intervals occasionally exhibited a bi-modal tendency. While these DSDs could be approximated as a gamma distribution, there were notable errors in the model, as well as a systematic error caused by the instruments [52]. On the other hand, the average DSDs differed significantly for medium- to larger-sized raindrops, indicating the spatial variability of DSDs and rain integral fields. Generally, the DSDs of the six stations are all approximately exponential shapes which are similar to the shape noted by Ulbrich et al. [53]. The mean concentrations of smaller and larger raindrops are higher at the Xuwen and Qingyuan stations, respectively. Research has shown that the topographical gradient has a clear impact on DSDs because small raindrops are more numerous at higher altitudes [54]. However, this is not apparent in our study area. Instead, the concentration of small raindrops is related more to distance from the sea and topography. Enping and Xuwen, which are in coastal areas, have the highest number of small raindrops, whereas Meixian and Luoding on the leeward side of inland mountains have the lowest number of small raindrops, which can be attributed to evaporation and modification by updraughts inducing a decrease in small raindrops [55]. Conversely, Qingyuan has the highest concentration of medium and large raindrops and also has larger raindrop diameters than other sites. This is indicative of a particular local microphysical process whereby the growth of raindrops is enhanced by the local topography through riming (at temperatures above 0 °C) and through coalescence below the melting layer [54,56].

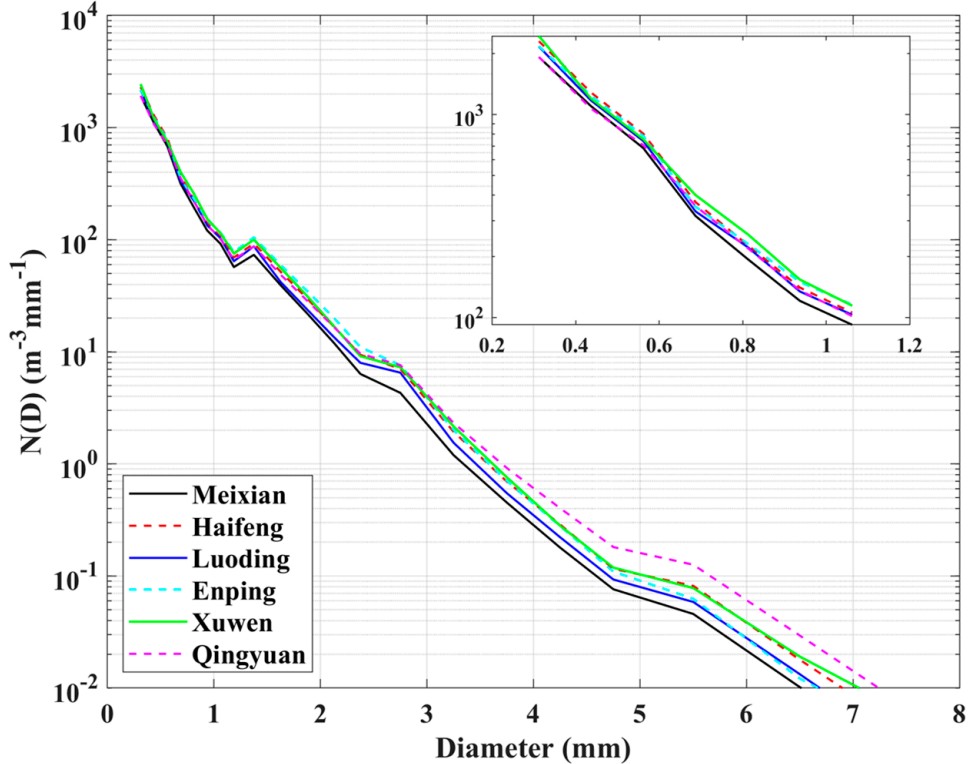

**Figure 7.** Variation of the mean raindrop concentration and raindrop diameter (Diameter, mm) in Meixian, Haifeng, Luoding, Enping, Xuwen and Qingyuan over five years.

### 3.3. DSD in Different Rainfall Rate Classes

In order to demonstrate the DSD differences in the six stations, the rainfall rates are divided into 12 rainfall rate classes, namely, C1: 0.1–0.2, C2: 0.2–0.4, C3: 0.4–0.7, C4: 0.7–1.0, C5: 1.0–2.0, C6: 2.0–5.0, C7: 5.0–8.0, C8: 8.0–12.0, C9: 12.0–18.0, C10: 18.0–25.0, C11: 25.0–40.0 and C12: >40 mm h$^{-1}$. These rainfall rate classes are chosen in such a way that the mean rainfall rate in each class is approximately the same in all the stations and the number of data points are sufficiently large in each class. A similar classification method was adopted by previous researchers [57,58]. Figure 8 shows the average variation in the number of raindrops per unit volume per diameter range with raindrop size for each rainfall rate class for the six stations. Essentially, the *N(D)* has an exponential shape which decreases monotonically for all rainfall rate types at the six observation sites and there is an increase in diameter as rainfall rates increased. When the rain intensity is less than 1.0 mm h$^{-1}$, there was no obvious difference in the *N(D)* between the sites. However, when the rainfall rate exceeded 1.0 mm h$^{-1}$, the differences in the *N(D)* at the six sites for all raindrops are relatively more obvious. However, as the rainfall rate increased, the *N(D)* of sites on the windward side, i.e., Haifeng, Enping and Qingyuan, exceed that of Meixian, Luoding and Xuwen which are on the leeward side. In particular, the Qingyuan site records a higher *N(D)* value than the other five sites for larger raindrops at rainfall rates above 1.0 mm h$^{-1}$.

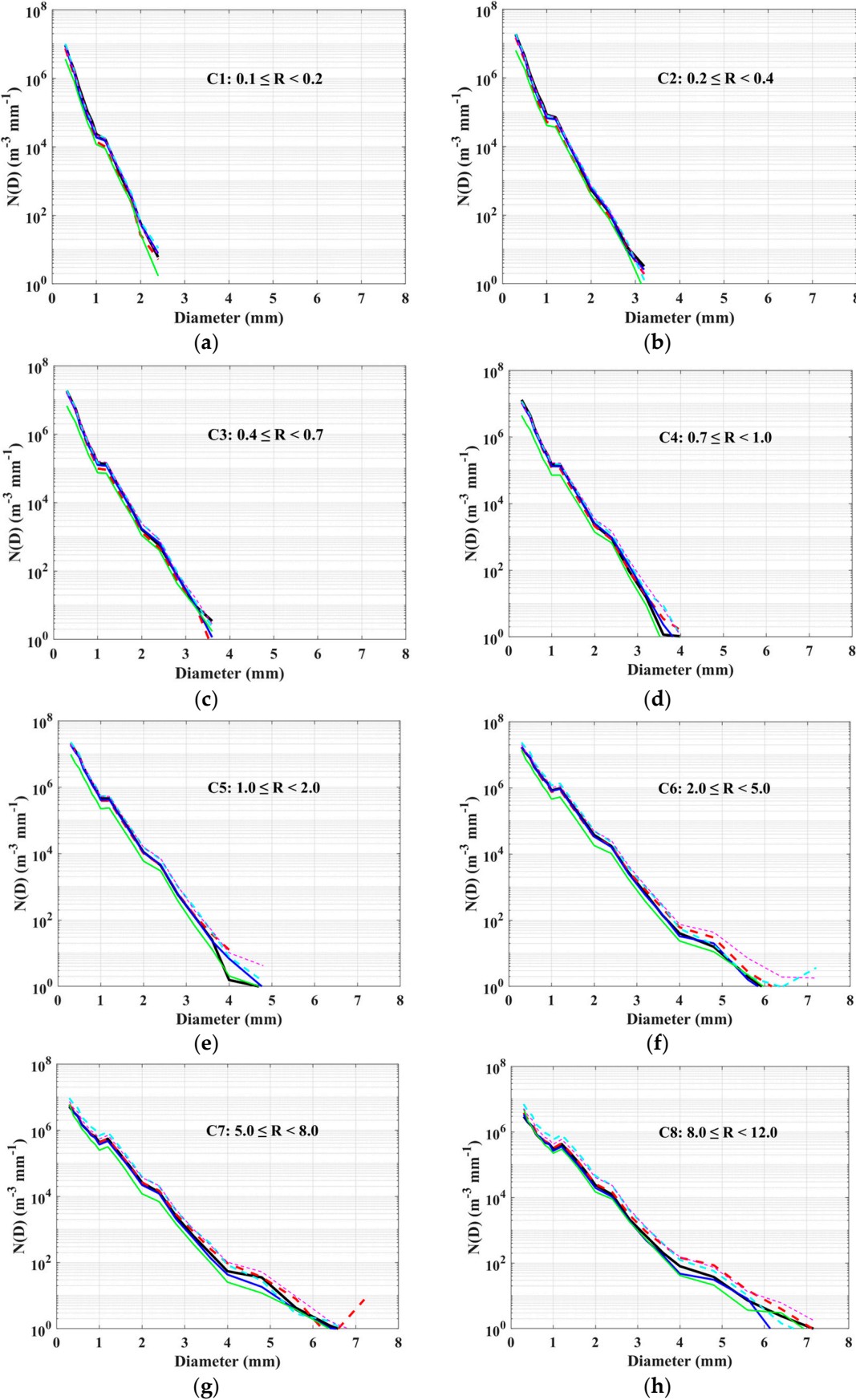

**Figure 8.** *Cont.*

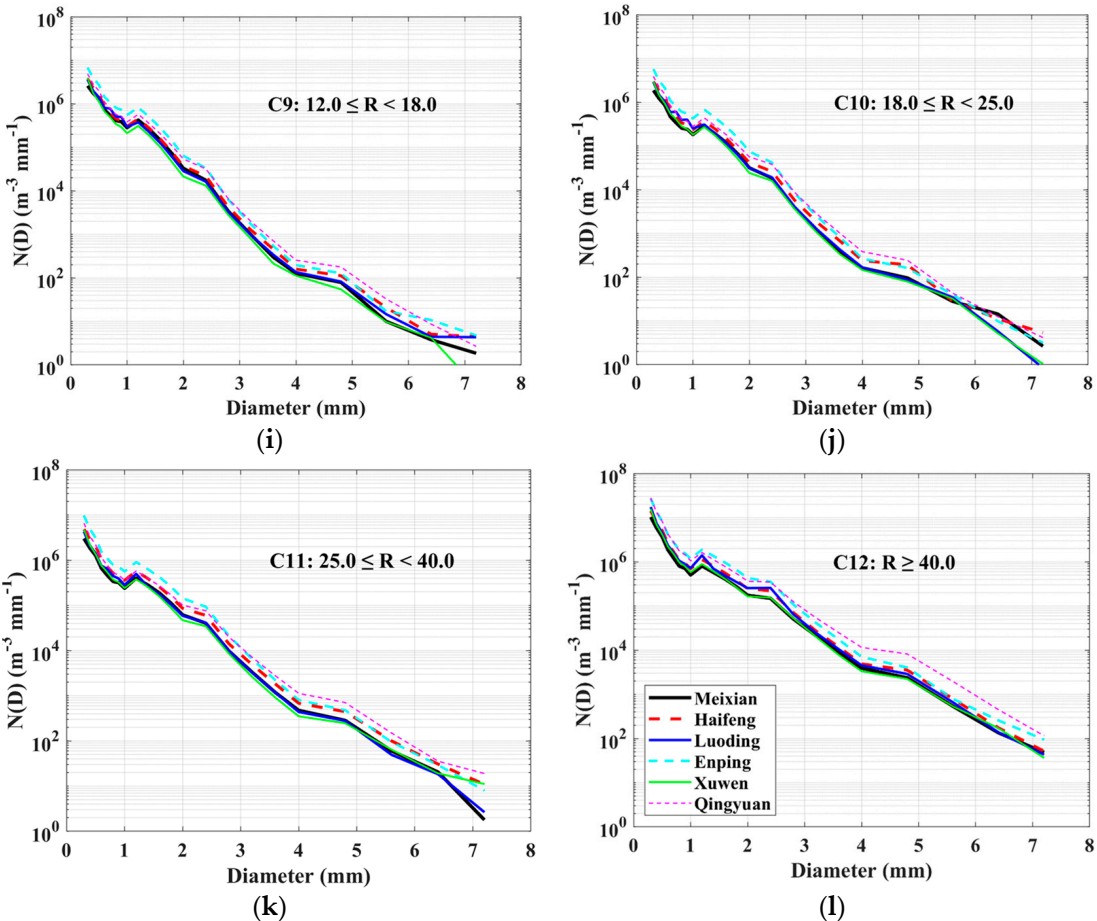

**Figure 8.** Average raindrop spectra of Meixian, Haifeng, Luoding, Enping, Xuwen and Qingyuan for rainfall in 12 rainfall rate classes (**a**) C1: 0.1–0.2, (**b**) C2: 0.2–0.4, (**c**) C3: 0.4–0.7, (**d**) C4: 0.7–1.0, (**e**) C5: 1.0–2.0, (**f**) C6: 2.0–5.0, (**g**) C7: 5.0–8.0, (**h**) C8: 8.0–12.0, (**i**) C9: 12.0–18.0, (**j**) C10: 18.0–25.0, (**k**) C11: 25.0–40.0 and (**l**) C12: >40 mm h$^{-1}$.

To investigate DSD variations in 12 rainfall rate classes at the six sites, the corresponding average $D_m$ and $\log_{10}N_w$ for the 12 rainfall rate classes are shown in Figure 9. For all six sites, $D_m$ was lower at a lower rainfall rate intensity, and gradually increased as the rainfall rate intensity increased. At lower rainfall rate intensities (C1, C2, C3 and C4), Xuwen had a $D_m$ that was a little higher than the other five stations. When the rainfall rate exceeded 1.0 mm h$^{-1}$, the $D_m$ of Qingyuan gradually became the largest of the six sites. The $\log_{10}N_w$ of the six sites shows the opposite pattern to $D_m$ when the rainfall intensity was below 5 mm h$^{-1}$, in that it steadily increased when the rainfall intensity exceeded 5 mm h$^{-1}$, with the exception of Xuwen, where it increased from 1.0 mm h$^{-1}$. At lower rainfall rates, an increasing $D_m$ and decreasing $\log_{10}N_w$ together with an increasing rainfall rate indicate that the enhanced collision–coalescence process is reducing the concentration of raindrops, contributing to the increase in the $D_m$ value. However, the $D_m$ and $\log_{10}N_w$ rose as the rainfall rate increased, reaching an equilibrium status through collision–coalescence and breakup between raindrops [59].

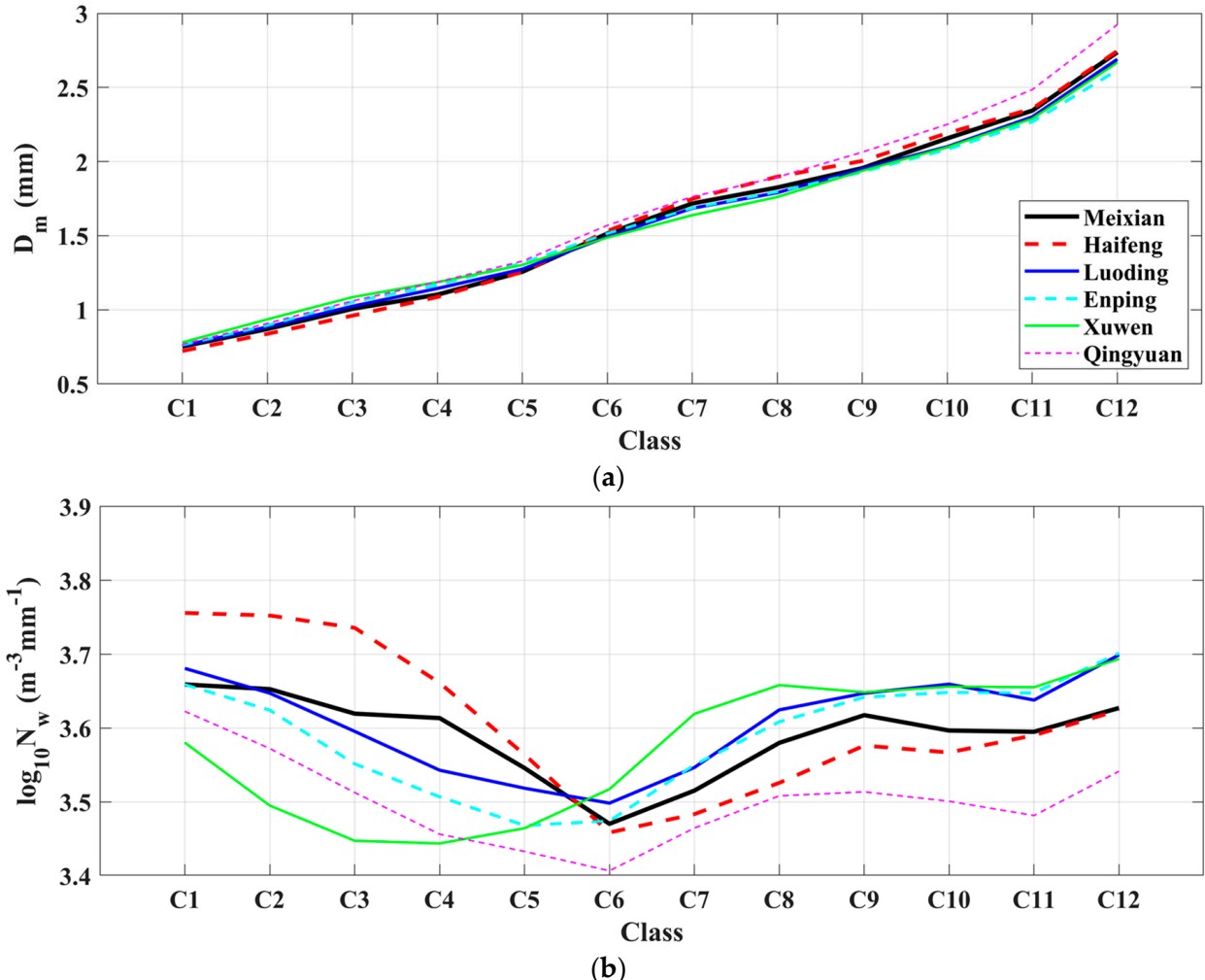

**Figure 9.** Distribution of (**a**) $D_m$ and (**b**) $\log_{10}N_w$ for rainfall at Meixian, Haifeng, Luoding, Enping, Xuwen and Qingyuan in relation to rainfall rates (C1: 0.1–0.2, C2: 0.2–0.4, C3: 0.4–0.7, C4: 0.7–1.0, C5: 1.0–2.0, C6: 2.0–5.0, C7: 5.0–8.0, C8: 8.0–12.0, C9: 12.0–18.0, C10: 18.0–25.0, C11: 25.0–40.0 and C12: >40 mm h$^{-1}$).

### 3.4. Distributions of $D_m$ and $N_w$

Figure 10 demonstrates a clear trend where $D_m$ and $\log_{10}N_w$ eventually reach an equilibrium state. Previous studies, such as Hu and Srivastava [60], have shown that for higher rainfall rates, the DSD reaches an equilibrium state where the coalescence and breakup of raindrops are nearly balanced. In this equilibrium state, $D_m$ typically remains constant and is independent of the rainfall rate. Any increase in rain intensity is mainly due to the variation in $N_w$, as indicated by studies such as Bringi et al. [41]. Based on Figure 9b, it can be observed that Dm tends to stabilize around a value of 2.5 mm for rainfall rates exceeding 100 mm h$^{-1}$, indicating that the DSDs have reached an equilibrium state. Therefore, it can be concluded that the DSDs reach an equilibrium state at rainfall rates above 100 mm h$^{-1}$.

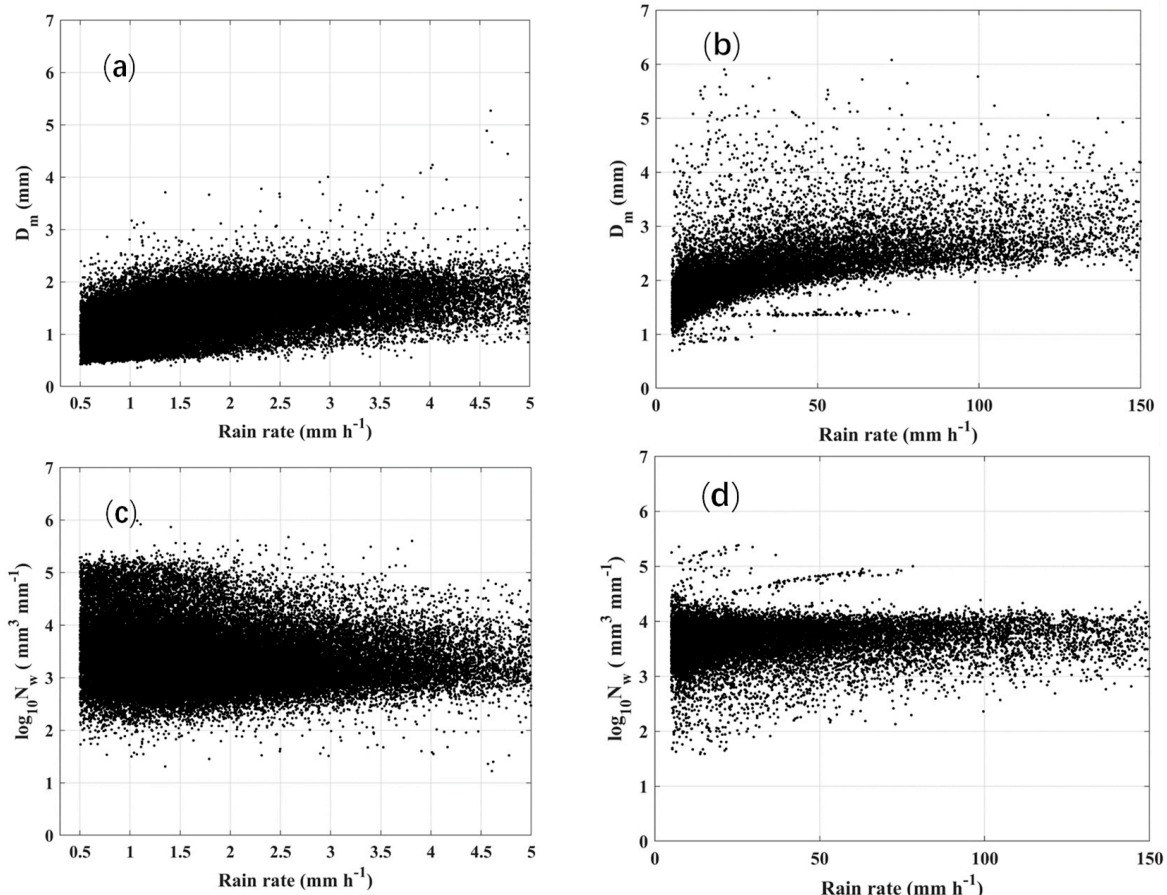

**Figure 10.** The scatter plots of $D_m$ vs. rain rate for (**a**) stratiform and (**b**) convective precipitation and $\log_{10}N_w$ vs. rain rate for (**c**) stratiform and (**d**) convective precipitation.

### 3.5. Stratiform and Convective DSDs

Rainfall can be classified in various ways because its microphysical properties depend on the rain type. Thus, a distinction has long been drawn between stratiform and convective types of precipitation. Convective precipitation is associated with small-scale convection and typically produces raindrops through the process of collision–coalescence and breakup processes, whereas stratiform precipitation is associated with the large-scale lifting of air and typically produces raindrops through the aggregation of snowflakes [61,62]. This section examines the differences in DSD between stratiform and convective rain types. The average raindrop spectra of stratiform and convective precipitation at the Meixian, Haifeng, Luoding, Enping, Xuwen and Qingyuan stations are shown in Figure 11. There are notable differences in stratiform and convective precipitation with narrower $N(D)$ distribution of lower concentrations of smaller raindrops and wider $N(D)$ distribution of higher concentrations of larger raindrops. The average $N(D)$ of stratiform precipitation at the six sites exhibits similar variations whereas the $N(D)$ convective precipitation at the six locations shows slightly different variability, in particular for medium- to large-sized raindrops. Luoding and Qingyuan have the most abundant medium- to large-sized raindrops.

Figure 12 shows the fitted power–law relationship of radar reflectivity Z and rainfall rate R of two weather types for the six different stations. Generally, the R increases with increasing Z for both precipitation types. In addition, the coefficient A and exponent b of the Z–R relationship ($Z = aR^b$) vary according to systematic error, synoptic weather situation, rainfall type and even instrument type [63], and are highly affected by DSD variability. Coefficient A represents the concentration of different raindrop sizes and exponent b represents the microphysical process. When exponent b is greater than unity and equal

to unity, it indicates the collision–coalescence and breakup processes, respectively [64]. Furthermore, for heavy (light) precipitation, the larger (smaller) coefficient and smaller (larger) exponent are expected [65]. The *Z*–*R* relationship of stratiform and convective precipitation for the six stations are obtained using linear regression of *Z* and *R* and are presented in Table 3. Both stratiform and convective precipitation have an exponent greater than unity value indicating that the size-controlled microphysical process is dominant in Guangdong Province. The stratiform precipitation at all six sites has similar *Z*–*R* relationships, whereas convective precipitation shows an appreciable variability between sites on the leeward (i.e., Meixian, Luoding and Xuwen) and windward sides (Haifeng and Enping, but not Qingyuan). The *R* is greater in the windward sites (with the exception of Qingyuan) compared with that of leeward sites with the same *Z*. Accordingly, the use of a single *Z*–*R* relationship will inevitably underestimate the *R* at one site and overestimate the *R* at the other site, indicating the importance of using specific *Z*–*R* relationships depending on different rain types, locations, seasons and topography. On the other hand, for convective rainfall, Fujiwara [66] proposed the large *A* (300–1000) and moderate *b* (1.25–1.65) for thunderstorms, and a smaller *A* and larger *b* (1.2–2.0) for continuous rain. It appears that the precipitation in Meixian, Luoding, Xuwen and Qingyuan belongs more to the thunderstorm type whereas that in Haifeng and Enping, which are coastal areas, is more of the continuous rain type. In addition, the equations for thunderstorms (A = 450 and b = 1.46) and continuous rainfall (A = 205 and b = 1.48) based on Fujiwara [66] give similar rainfall rate calculation results for Meixian, Luoding, Xuwen and Qingyuan but showed slight differences for Haifeng and Enping. The Fujiwara relationship [66] for continuous rainfall overestimates the rainfall rate for convective precipitation when the rainfall rate exceeds 40 mm/h. These features are less pronounced than they are in the case of stratiform precipitation, possibly because of the small scale of convective precipitation, less of which passes through the observation station compared with stratiform precipitation [67].

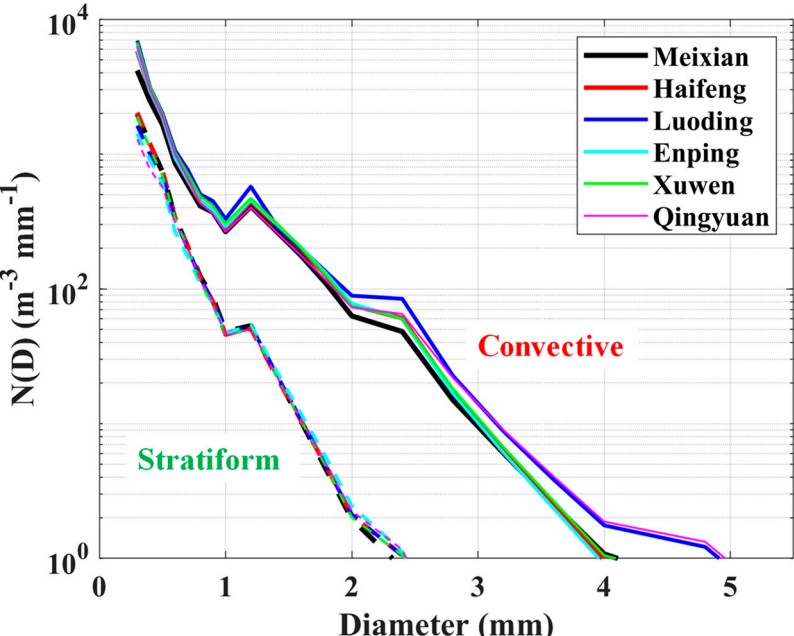

**Figure 11.** Average raindrop spectra of stratiform and convective precipitation at Meixian, Haifeng, Luoding, Enping, Xuwen and Qingyuan stations. the dashed line represents the stratiform precipitation and the solid line represents the convective precipitation.

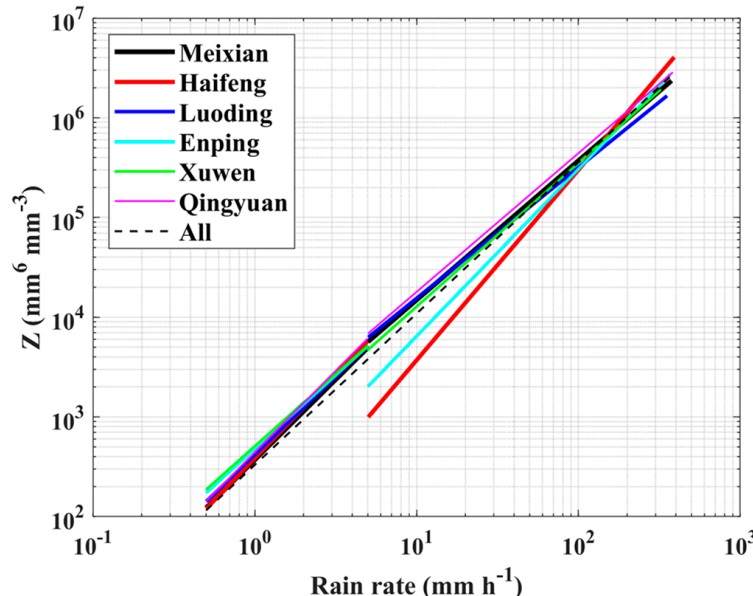

**Figure 12.** Radar reflectivity $Z$ and rainfall rate $R$ relationship of stratiform and convective precipitation at Meixian, Haifeng, Luoding, Enping, Xuwen, Qingyuan stations and for all data.

**Table 3.** Relationship between radar reflectivity $Z$ and rainfall rate $R$ for the data in Figure 12.

| Site | Z–R Relationship | | |
|---|---|---|---|
| | **Stratiform** | **Convective** | **All Rainfall** |
| Meixian | $Z = 372R^{1.6}$ | $Z = 599R^{1.4}$ | $Z = 574R^{1.4}$ |
| Haifeng | $Z = 385R^{1.7}$ | $Z = 46R^{1.9}$ | $Z = 285R^{1.6}$ |
| Luoding | $Z = 418R^{1.6}$ | $Z = 756R^{1.3}$ | $Z = 915R^{1.3}$ |
| Enping | $Z = 480R^{1.5}$ | $Z = 136R^{1.7}$ | $Z = 260R^{1.6}$ |
| Xuwen | $Z = 508R^{1.5}$ | $Z = 462R^{1.4}$ | $Z = 242R^{1.6}$ |
| Qingyuan | $Z = 441R^{1.6}$ | $Z = 730R^{1.4}$ | $Z = 776R^{1.4}$ |
| All | | $Z = 330.65R^{1.5}$ | |

## 4. Summary and Conclusions

This study addressed the statistical characteristics of regional variability over complex terrain in southern China using disdrometer observations from 1 March 2018 to 30 August 2022 (~5 years) collected by six stations, namely, the Meixian, Haifeng, Luoding, Enping, Xuwen and Qingyuan sites. The data were separated into stratiform and convective precipitation, and the precipitation was divided into 12 classes based on the rainfall rate. The main conclusions can be summarized as follows:

(1) The median $D_m$ value was higher on the windward than on the leeward side, and the windward-side stations also showed greater $D_m$ variability. With regard to the $N_w$ value, the median $N_w$ value decreased from coastal to mountainous areas. This variation in $N_w$ can be explained by the distance to the ocean, which shows the importance of terrain on the rainfall in a local area.

(2) Although there were some differences in $D_m$, $\log_{10}N_w$ and *LWC* frequency between the six stations, there was still a degree of similarity. The $D_m$, $\log_{10}N_w$ and *LWC* frequency all showed a single-peak curve with a highest frequency of 15% when the $D_m$ value was 1.0 mm at Meixian and a highest frequency at 6% when the $\log_{10}N_w$ value was 3.6 mm at Qingyuan. Furthermore, the Meixian and Luoding areas, which are far from the ocean and on the leeward side of mountains, had a similar *LWC* frequency trend, whereas Haifeng, Enping, Xuwen and Qingyuan, which are near the ocean or on the windward side of mountains, were similar to each other. Despite some minimal similarities, the DSD

distribution at coastal sites and inland sites remained different, indicating the complicated cloud microphysical processes in these six regions.

(3) The diurnal variation of the mean $\log_{10}N_w$ had a negative relationship with the $D_m$ diurnal variation, although the inverse relationship was not very notable at the Haifeng site. The $\log_{10}N_w$ had its minimum value in the late afternoon, probably because of the presence of precipitating convective clouds at this time, and the diurnal mean rainfall rate also showed a peak in the afternoon which exceeded the maximum at night, indicating that strong land heating during the day significantly influenced local DSD variation.

(4) The $N(D)$ had an exponential shape which decreased monotonically for all rainfall rate types at the six observation sites and an increase in diameter due to an increase in rainfall rate was also noticeable. As the rainfall rate increased, the $N(D)$ of sites on the windward sides (i.e., Haifeng, Enping and Qingyuan) exceeded that of sites on the leeward side (i.e., Meixian, Luoding and Xuwen) and the difference between them also became distinct. At lower rainfall rates, increasing $D_m$ and decreasing $\log_{10}N_w$ together with an increasing rainfall rate indicated that the enhanced collision–coalescence process was reducing the concentration of raindrops and contributing to the increase in the Dm value. On the other hand, at higher rainfall rates, the $D_m$ and $\log_{10}N_w$ increased in line with the increase in rainfall rate reaching an equilibrium status as a result of collision–coalescence and breakup between raindrops.

(5) The characteristic differences of DSD in the six stations described above also revealed appreciable variability in convective precipitation between the sites on the leeward side (i.e., Meixian, Luoding and Xuwen) and those on the windward side (Haifeng and Enping, but not Qingyuan). Accordingly, the use of a single *Z–R* relationship will inevitably underestimate R at one site and overestimate R at the other site, showing the importance of using specific *Z–R* relationships depending on different types of rain, locations, seasons and topography.

This study only focused on observational data obtained from the six different locations. Future studies are still vital to understand other locations using disdrometer data, not only in terms of terrain influence, but also considering the weather systems and environmental conditions in Guangdong Province, thereby providing a more complete assessment of the microphysical statistical properties of this region. Despite the valuable insights provided by the DSD spectra, it is important to acknowledge the limitations of the HY-P1000 disdrometer in accuracy measuring raindrops. To address this issue, future studies could incorporate multiple instruments, such as the two-dimensional video disdrometer in the above area. Additionally, combining data from other observation methods, such as the vertically pointing profiler radar, could improve the classification of different types of rainfall. Thus, it is recommended that future studies consider these factors to enhance their results. Meanwhile, it is also necessary to study the raindrop spectra on windward and leeward slopes using individual cases as examples to enhance our understanding of microphysical variations on mountain areas and also offer insights for improving local precipitation forecasting models.

**Author Contributions:** Conceptualization, C.C.; writing—original draft preparation, A.Z.; writing—review and editing, A.Z. and L.W. All authors have read and agreed to the published version of the manuscript.

**Funding:** This research was funded by "National Natural Science Foundation of China, grant number 41975138", "Natural Science Foundation of Guangdong Province, grant number 2022A1515011814", "The Open Grants of the State Key Laboratory of Severe Weather, grant number 2022LASW-B15", "Radar Application and Short-term Severe-weather Predictions and Warnings Technology Program, grant number GRMCTD202002", "Science Technology Research Program of Guangdong Meteorological Service, grant number GRMC2020Z03" and "Water Resource Science and Technology Innovation Program of Guangdong Province, grant number 2022-02".

**Informed Consent Statement:** Written informed consent has been obtained from the patient(s) to publish this paper.

**Data Availability Statement:** The data are not publicly available due to restrictions privacy.

**Acknowledgments:** Thanks are given to Yunce Liu who gave us constructive advice on this paper. Further, we really appreciate the three anonymous reviewers for their great efforts to review this paper and give comments to improve the manuscript.

**Conflicts of Interest:** The authors declare no conflict of interest.

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
