# Peer review of "Regional Variability of Raindrop Size Distribution from a Network of Disdrometers over Complex Terrain in Southern China"

_remotesensing, doi:10.3390/rs15102678_

Round 1
Reviewer 1 Report (Previous Reviewer 1)
The paper provide a good insight of the precipitation measured by disdrometers during the period 1 March 2018 to 30 August 2022.
The paper is in good shape but it requires a minor review.
The authors didn’t mention the measurement limitations of the HY-P1000 disdrometer in the third paragraph of section 2.2, and please add it.
Author Response
Respond: Thanks for the advice. Actually, there is limited research on the HY-P1000 disdrometer, and the company itself only provides Chinese manual which is not very formal. The principle of HY-P1000 disdrometer is the same as OTT disdrometer, therefore, we add OTT limitations in Line 138-146.

Reviewer 2 Report (New Reviewer)
Please see my comment in the attached file.

Round 2
Reviewer 2 Report (New Reviewer)
In general, the author has revised the manuscript as suggested, but there are still several things that have not been answered adequately, including:
1. In the introduction, why the leeward side's raindrop size distribution is essential to study still needs to be explained. An explanation instead of raindrops on the ocean and land.
2. The author agrees that instrument errors cause multiple peaks of raindrop spectra. This needs to be explained better and supported by adequate references.
3. In Table 3, adding one column for the Z-R relationship would be better without grouping (all) for each location, not just "all" for all locations.
4. Several of our comments were only answered as a next step (future research), but this was not found at the end of the conclusion as future research.
Author Response
Please see the attachment.

This manuscript is a resubmission of an earlier submission. The following is a list of the peer review reports and author responses from that submission.
Round 1
Reviewer 1 Report
This paper analyze the climatological DSD characteristics over complex topography from the disdrometer measurements during five-year period in six locations at the south of China. This paper is written in a direct and organized way and it’s generally easy to read which can be publishable in Remote Sensing despite some points which are mentioned below:
Major concerns:
1. the authors should note the limitation of the HY-P1000 disdrometer instruments which could affect the results of this paper. In addition, after the type name, i.e., HY-P1000, is indicated at the first occurrence, it can be completely defaulted later, and just represented by disdrometer because of all the same.
2. The discussion is mostly carried out in the results section, while the section of Discussion and Conclusions has nearly no discussion and it contains instead a detailed summary of the results. I advise to reorganize the three parts (results, discussion and conclusion) in a clear way.
The remaining points are mostly language concerns with some suggestions:
L18: The font format of full-text variables should be unified and all italics should be used;
L18: than that in…;
L21, 22: LWC first appears in L21;
L25: The diurnal of mean rain rate also showed peak in the afternoon… [showed a peak]
Line 27: The N(D) showed exponential shape which… [showed an exponential]
L90-94: Guangdong is one of the maximum region-averaged … while located at the leeward side, Meixian Luoding and Xuwen experience 1000~1500 mm. [ the author should rewrite the above two sentences to make it more reasonable.]
L92: the Guangdong Province, “the” should be removed;
L132: “th” should be superscript, and check the full text;
L134: 139: LWC has been defined previously;
L140: spectra should be removed;
L158: the reference classification method should be explicitly listed;
L162: RSD?
L173: rain spectra, it is recommended to use DSD uniformly in the full text;
L188: Enping, Xuwen and Qingyuan shows a bigger value [“show” instead of “shows”]
L202: Xuwen which has a less yearly rainfall amount than the Haifeng, Enping and Qingyuan which is the… [“have” instead of “has”]
L220: log10Nw larger 3.7 and smaller than 4.3. [log10Nw is larger 3.7]
L231: the colons in the middle of time format should be removed in full text;
L240: All site except… [“All sites” instead of “All site”]
L259: …is a influence…[“an” instead of “a”]
L283: (a) in Figure 6.

Reviewer 2 Report
The manuscript deals with the regional variability of RDSD using the disdrometers network consisting of 6 sites over complex terrain in southern China. I think that the authors organized the manuscript very well and they analyzed the DSDs characteristics including Z-R realtions using long period data set. However, I think that the authors need to resolve some issues before publication.
Comments
1. Line 117: The authors applied the quality control of disdrometer data. For quality control of disdrometer like Parsivel, we usually use the method proposed by Friedirich et al.(2003). I would like to know that the authors would explain the difference between their method and the method used in this study.
2. Figure 2: I think that the authors had better describe the data set used for comparison the rainrate. How many sites are used for this comparison?
3. Line 204: log10Nw, 10 should be subscript.
4. All figures, (a), (b), etc. should be located upper part of the figures.
5. Figure 4: I think that the authors need to describe the bin size of each variable in the manuscript.
6. Figure 6: I could not recognize the difference of number concentration at small size drop like smaller than 1 mm. For better understanding, the authors need to zoom in them.
7. Lines 289-290: The interval of rainrate categories were too many to understand. From Figure 8, the difference of Dm is clearly shown at the point of C6. If it is possible, I would like to recommend that the authors would reduce the number of the category.
8. Title of Section 4: I think that there are no discussions in this section. Therefore, the title of Section 4 would be changed by Summaries and Conclusions or Conclusions.
Reviewer 3 Report
This paper analyses a multi-year disdrometer data set in a region with orography. The analysis of the data appears sound, however the analysis of the results with respect to the meteorology (dynamic, synoptic, local forcing of the flow and the resulting precipitation process) is in my view lacking. An introduction on the expected microphysical processes of the precipitation process typical in this region is missing. You should try to make a better link to any work that is relevant to yours in order provide some context for the readers.
The Z/R relationships have to be compared to existing relationships. That is completely missing.
you don’t show any inter-annual results. No variability present?
The text needs heavy English editing (grammar, typos)
L. 78: five years of disdrometer data is a long data set, but does not provide a climatological examination (at least > 10 years of data are needed). You can perhaps provide an assessment on the year to year variability
L 83 ff: so in which climatological zone are we know? Make a statement here
l. 165: did you verify the separation into convective and stratiform regime with some independent data? Radar data?
L169: why is LWC crucial for ice phase AND the warm rain process? What do want to say here?
L 199: so what’s the physical process explaining the decrease of Nw towards the mountain. I trust your analysis, but what are the physical mechanisms behind that observation?
L 235: the diurnal variation seen in the coastal region: so a precipitation system seen in the mountains will eventually make to the coast? You seem to suggest this when you say that “complex terrain…influence the microphysical process. Or do you want to express something else here: What about land-see circulations that induce diurnal precipitation patterns? Or is the precipitation mainly governed by the synoptic (large scale) forcing?
What about any seasonal variability? Is there any?
L 328: statement that cloud droplets grow by vapor diffusion in stratiform precipitation seems wrong unless you really consider tropical rain process. So please make clear statement here because in mid-latitude the stratiform precipitation process always involve mixed-phased processes! So you don’t see any bright band in radar data for the stratiform cases??
L 362: did you compare the Z/R relationships with relationships published in literature? Is there agreement?
L 372ff: the physical explanation is missing.
L 399ff: single Z/R relationship not working: trivial statement I would say. It is state of the art to use Z/R relationship depending on the precipitation process. What is the Chinese meteorological service doing operationally? I believe you have polarimetric radar systems as well which should provide even more information on the precipitation processes. Can you provide some information?
L 405: as for the future: I would suggest to connect the results to radar data, if available. Furthermore, I would try go more into the physical processes and the relation to the large scale forcing….but you seem to go into this direction.
Round 2
Reviewer 3 Report
Thank you for the revision. The language improved quite a bit, but you should use present tense when reporting about your results. You attempted to introduce more physical explanations. This is acknowledged, but it is often wrong (see examples below) Unless I have missed it, your discussion on diurnal variation is missing a statement on possible seasonal effects, and the variability at a given time of the day.
In the current state, this paper is not ready to be published.
l.34 in the abstract you report about the DSD differences, but how does this improve the understanding in microphyiscs? Remove this statement or specifically write about the improved understanding in microphysics.
l 58: there also studies outside of China. .. it sounds like there has been only work in China.
l 73: longterm of drop size distributions (DSDS)..... ARE still fairly limited.. I would say, or?
l 77: ... properties of cloud droplets and precipitation: what are properties of precipitation? you refer to type (solid or liquid)? Or what is meant here? Sentence is too general and therefore misleading
l 85: here you should state somewhere whether we deal with mix-phased precipitation or not
l. 91: which cloud microphyiscal processes? Give an example? or rewrite
l 97: Units are missing (height in m); please use si units (km instead of miles). Indicate the disdrometer site names.
l. 111: you mention site names which cannot be found in the map (fig 1)
l 112...is this height difference significant???? I would say no.. More important should the location with respect to the main flow.
l 118: ...the sample size IS....
l 119ff use present tense since you list here characteristics of the instrument which do not change, or??
l 123 8 mm/h...
l 125 ...used for calculation ...of what?
l 127: ...understand... I think you characterize the DSD with standard methods (which is fine), but you don't provide any insight into the mechanisms of the precipitation which lead to the observed DSD. So please rewrite or prove me that you actually contribute to the understanding.
l 152 please use present tense
l. 159 ff as an example in general, if you report about the results you should use present tense.
L 177: the reference you cite is about precipitation that evolves with ice phase… and here you talk about a warm rain process (which does not involve the ice phase). Wrong citation? Please clarify. Is the LWC value really such it supports the physical explanation you attempt to give? Sorry, sounds wrong to me, unless you provide some proof from radar data for example. Is the Luoding site special with respect to the siting (lee/luv e.g.). An interpretation with respect to orography is missing.
L 208 so what is the physical explanation for Nw as a function of the distance to the ocean?
L 234: only because you see some differences doesn’t mean that the microphysical processes are “complicated”
l. 239: the discussion on diurnal variations has to be reworked, see the next comment.
L 272: Diurnal variation: what is the variability (the spread of the retrieved parameters) as function of time. There are no seasonal variations? Is the variation identical if you take ony data from say DJF versus JJA (June, July, August)?
L 293. Riming is a process in the ice phase!
L 340 ff wrong explanation of ice particle growth. Suggest to check some text books on this subject. Aggregation, riming happens also in stratiform situations.
L 374ff: so the Fujiwara Z/R relationships agree with you finding or not?? Make a clear statement please.